# Is the Combination of Robot-Assisted Therapy and Transcranial Direct Current Stimulation Useful for Upper Limb Motor Recovery? A Systematic Review with Meta-Analysis

**DOI:** 10.3390/healthcare12030337

**Published:** 2024-01-29

**Authors:** Juan J. Bernal-Jiménez, Begoña Polonio-López, Ancor Sanz-García, José L. Martín-Conty, Alfredo Lerín-Calvo, Antonio Segura-Fragoso, Francisco Martín-Rodríguez, Pablo A. Cantero-Garlito, Ana-Isabel Corregidor-Sánchez, Laura Mordillo-Mateos

**Affiliations:** 1Faculty of Health Sciences, University of Castilla-La Mancha, 45600 Talavera de la Reina, Spain; juanjose.bernal@uclm.es (J.J.B.-J.); ancor.sanz@uclm.es (A.S.-G.); joseluis.martinconty@uclm.es (J.L.M.-C.); asegurafr@gmail.com (A.S.-F.); pablo.cantero@uclm.es (P.A.C.-G.); anaisabel.corregidor@uclm.es (A.-I.C.-S.); laura.mordillo@uclm.es (L.M.-M.); 2Technological Innovation Applied to Health Research Group (ITAS Group), Faculty of Health Sciences, University of de Castilla-La Mancha, 45600 Talavera de la Reina, Spain; 3Neruon Neurobotic S.L., 28015 Madrid, Spain; alerin@neuronrehab.es; 4Department of Physiotherapy, Faculty of Health Sciences, University La Salle, 28023 Madrid, Spain; 5Faculty of Medicine, University of Valladolid, 47005 Valladolid, Spain; fmartin@saludcastillayleon.es; 6Advanced Life Support, Emergency Medical Services (SACYL), 47007 Valladolid, Spain

**Keywords:** robot-assisted therapy (RAT), transcranial direct current stimulation (tDCS), stroke, upper limb recovery, hand dexterity, spasticity

## Abstract

Stroke is the third leading cause of disability in the world, and effective rehabilitation is needed to improve lost functionality post-stroke. In this regard, robot-assisted therapy (RAT) and transcranial direct current stimulation (tDCS) are promising rehabilitative approaches that have been shown to be effective in motor recovery. In the past decade, they have been combined to study whether their combination produces adjuvant and greater effects on stroke recovery. The aim of this study was to estimate the effectiveness of the combined use of RATs and tDCS in the motor recovery of the upper extremities after stroke. After reviewing 227 studies, we included nine randomised clinical trials (RCTs) in this study. We analysed the methodological quality of all nine RCTs in the meta-analysis. The analysed outcomes were deficit severity, hand dexterity, spasticity, and activity. The addition of tDCS to RAT produced a negligible additional benefit on the effects of upper limb function (SMD −0.09, 95% CI −0.31 to 0.12), hand dexterity (SMD 0.12, 95% CI −0.22 to 0.46), spasticity (SMD 0.04, 95% CI −0.24 to 0.32), and activity (SMD 0.66, 95% CI −1.82 to 3.14). There is no evidence of an additional effect when adding tDCS to RAT for upper limb recovery after stroke. Combining tDCS with RAT does not improve upper limb motor function, spasticity, and/or hand dexterity. Future research should focus on the use of RAT protocols in which the patient is given an active role, focusing on the intensity and dosage, and determining how certain variables influence the success of RAT.

## 1. Introduction

Stroke is the third leading cause of disability in the world due to brain tissue damage following ischemic or haemorrhagic lesions [1]. Survivors present problems carrying out basic activities of daily living (BADLs) and instrumental activities of daily living (IADLs) [2], and their perception of their quality of life decreases. Hemiplegia and upper limb weakness are common sequelae after stroke and are among the most important causes of -limitations in the independence and quality of life of stroke survivors [3]. Spasticity is another symptom that has a major impact on BADLs and quality of life in patients who underwent stroke [4].

Although physical therapies improve some motor impairments [5], sequelae are still present. Among the techniques with the greatest scientific support, robot-assisted therapy (RAT) [6,7] and non-invasive brain stimulation (NIBS) [8,9] are the predominant ones. Both, when used together, provide relevant improvements in postinjury recovery; RAT is a good tool to incorporate important concepts, such as intensity and repetition, in rehabilitation [10,11], while NIBS is a great strategy to address the interhemispheric rivalry that occurs after stroke [12].

As far as motor disorders are concerned, hand problems cause part of this disability, generating a high cost, both at the personal and social levels. Moreover, conventional rehabilitation is a time-consuming process, making RAT particularly interesting [13,14]. A Cochrane systematic review based on 45 trials with a total of 1619 patients found benefits in the development of activities of daily living and in the recovery of motor function in stroke survivors who received RAT. However, the results should be interpreted with caution, since there is a high degree of variability in the trials in terms of intensity, dose, or characteristics of the participants [15], which reflects the necessity for further research. The intensity and dose, i.e., the amount of training or the number of repetitions performed, are very relevant since they are critical for the activation of the motor learning process, having a decisive influence on its effectiveness. Although the number of repetitions necessary to facilitate plasticity has not been defined, it has been shown that the amount of movement provided in rehabilitation is not sufficient [15]. In this sense, robotic devices are one of the best tools for physical and occupational therapists since they allow a greater amount of movement to be generated. This increase in the number of repetitions is crucial for plasticity and, therefore, for motor recovery after a stroke.

Another of the most common processes after a stroke is the interhemispheric competition, which indicates that the suppression of the excitability of the unaffected hemisphere reduces the interhemispheric inhibition of the affected hemisphere, thus improving alterations after the stroke [16,17]. NIBS is an excellent strategy to develop this model and improve symptoms after stroke. NIBS includes different techniques, such as transcranial magnetic stimulation (TMS) and transcranial direct current stimulation (tDCS). tDCS modulates cortical excitability through a galvanic electric current and can induce brain plasticity, with possible improvement being associated with this process [18]. tDCS has been combined with different techniques for recovery after stroke [9,19,20] and has been used in other pathologies [21,22]. One of the most widely explored combinations is RAT with tDCS. This combination seems to offer promising results for motor recovery, but nevertheless, the results of the studies did not show a clear difference compared to conventional treatment.

In the past decade, the combination of tDCS and RAT has been highly developed by several researchers who consider that the combination of both techniques generates an adjuvant effect and provides better results in motor recovery after stroke. Four systematic reviews have been performed to analyse the combination of tDCS with RAT for motor recovery of the upper limb after stroke [23,24,25,26]. Two of the reviews did not perform a meta-analysis [23,26]. Another study analysed the effects on the upper and lower limbs [24]. Reis (2021) performed a meta-analysis on activity limitations in a limited way and did not consider the type of activity performed in RAT or the number of repetitions and included other NIBS techniques [25].

None of the previous reviews analysed parameters such as the number of repetitions or the intensity of the RAT, which are fundamental aspects for defining the gold standard in the combined treatment of RAT with tDCS. Hence, we consider that there is a necessity for a review that addresses more variables to shed light on this critical field.

The main objective of this review is to determine the effectiveness of the combined use of RAT and tDCS in upper limb function motor recovery after stroke. The secondary objectives are to analyse how the combination of the techniques can improve motor function, manual dexterity, spasticity, and the performance of activities of daily living. In addition, the variables that may influence the application of these techniques are observed in order to describe the characteristics of the patients who may benefit the most.

## 2. Materials and Methods

The first step in conducting this systematic review was to ask the question according to the PICO format. The question raised was as follows: in patients who underwent stroke (P), what is the effect of combining robot-assisted therapy with transcranial direct current stimulation (I) compared to robot-assisted therapy alone (C) on the motor recovery of the upper limb (O)? The second step was to design the protocol. At this time, the task force, the inclusion and exclusion criteria, the target databases, and the search strategy were established. In this step, the protocol was registered at PROSPERO with the following code: CRD42022304888. The next step was to implement the search strategy in the selected databases. With the first results, a screening of the studies was carried out in two phases. In the first phase, the studies were screened based on titles and abstracts. In the second phase, the full text was screened. Once the final studies were selected, we moved on to a data extraction phase, which was carried out with an ad hoc archive. We assessed the methodological quality of the studies and analysed data from those studies. Figure 1 shows this process. This review was performed and reported according to the Preferred Reporting Items for Systematic Reviews and Meta-Analysis (PRISMA) statement [27].

### 2.1. Selection Criteria, Identification, and Selection of Studies

Studies with adults diagnosed with stroke were included. Patients suffering from more than one lesion were excluded from the study. Regarding the characteristics of the participants, sex, type of stroke, time of evolution, location of the lesion, mean age, and damaged hemisphere were noted.

The inclusion criterion for intervention was to combine tDCS with RAT on the upper limb. tDCS could be applied before, during, or after the RAT session. The electrodes had to be placed according to the 10/20 system. The RAT had to be repetitive, planned, and structured with the aim of improving upper limb functionality.

The frequency of sessions, duration of each session, duration of the program, tDCS parameters (timing, setup, intensity, and type of stimulation), and RAT parameters (e.g., type of device—end-effector or exoskeleton—and number of repetitions) were recorded to compare the similarity between studies. The experimental group had to receive RAT and real tDCS, and the control group had to receive the same type of RAT as the experimental group combined with sham tDCS, thus allowing for the assessment of the effect of tDCS combined with RAT.

A comprehensive literature search was performed to find relevant articles in four databases: PubMed, Web of Science (WOS), SCOPUS, and the Cochrane Library. The search string was adapted to each database. The search was performed from database inception to 1 January 2023. The search string included words related to stroke, upper limb robotic treatment, and transcranial direct current stimulation (search strategy is presented in Appendix A). This search string was set up by two independent investigators (LMM and JBJ) using the inclusion criteria shown in Appendix A. Both reviewers were blinded to the authors, journals, and study results. A third reviewer (BPL) resolved ambiguities and disagreements between the first two reviewers.

### 2.2. Quality Evaluation of Involved Studies

The methodological quality of the included studies was assessed following the criteria and scores of the PEDro Scale (Physiotherapy Evidence Database). The PEDro scale is an 11-item scale designed to rate the methodological quality (internal validity and statistical information) of randomised trials. Each item, except for Item 1, contributes 1 point to the total score (range of 0 to 10 points). The GRADE system was used to assess the quality of the evidence. The GRADE system takes into account eight factors to reduce or increase the level of evidence, namely, five for downgrading and three for upgrading.

The I^2^ statistic was calculated, which was interpreted as absent (0), low (25), moderate (50), or high (75 or higher). The chi-square test was used to assess whether the differences observed in the results were compatible with chance. If the total number of patients included in a systematic review is less than the number of patients generated by a conventional sample size calculation for a single adequately powered trial, consider the rating down for imprecision. Imprecision was assessed by the calculation of the optimal information size (OIS). The OIS is calculated using a conventional determination of the sample size needed to detect an SMD equal to the minimum clinically important difference using the post-intervention standard deviation of the control.

It was assessed using the funnel plot created with RevMan and complemented with the DOI plot created with METAXL. Egger’s method and Begg’s test with Epidat 3.1 and the Luis Furuyama-Kanamori index (LFK) were used. Begg’s and Egger’s tests contrast the null hypothesis of the absence of publication bias. Begg’s test uses rank correlation between the effect of the standardised intervention and its standard error. Egger’s test uses linear regression of the estimate of the effect of the intervention against its standard error, weighted by the inverse of the variance of the estimate of the effect of the intervention. An LFK index of <1 was considered non asymmetric; between 1 and 2 was considered minor asymmetry; and ˃2 was considered major asymmetry.

### 2.3. Outcome Measures

The first outcome measure was the severity of the deficit, measured with the Fugl-Meyer Assessment for Upper Extremity (FMA-UE). The second outcome measure was hand dexterity, measured with the Box and Block Test (BBT); followed by spasticity, measured using the Modified Ashworth Scale (MAS); and finally, activity, measured using the Barthel Index (BI).

The timing of the measurements and the procedure used were recorded to assess the appropriateness of combining the studies in a meta-analysis.

### 2.4. Data Extraction and Analysis

Data regarding sample size, participant characteristics, characteristics of the robotic intervention, characteristics of the tDCS intervention, outcomes, and conclusions were extracted from each selected article. These data were recorded by two independent investigators (LMM and JBJ) and verified by a third party (BPL). Data were recorded in tables designed for this purpose, according to the Cochrane Handbook for Systematic Reviews of Interventions. Full details of the data extraction can be found in Appendix A.

Outcome measures were collected at baseline and at the first follow-up. Post-intervention differences were calculated for the experimental and control groups. The deviations (SD) of these differences were calculated by imputing a correlation coefficient that was calculated in studies with sufficient information using the pre- and post-SD and the SD of the difference. The mean of these coefficients (r = 0.85) was calculated and applied to the other studies.

The effect size was measured using Hedges’ G (adjusted standardised mean difference (SMD)) with a 95% confidence interval (95% CI). The overall effect size of the set of studies, weighted by the sample size of each study, was calculated using the inverse variance method and a random-effects model. Its 95% CI and statistical significance were calculated using the Z test. The overall effect size was interpreted using Cohen’s criteria for pooled estimates [28]; SMD > 0.20, small effect; SMD > 0.50, medium effect; and SMD > 0.8, large effect.

## 3. Results

### 3.1. Flow of Studies through the Review

The COVIDENCE program was used to analyse the articles [29]. A total of 227 related articles were found. After eliminating duplicate studies (n = 97), a first screening was performed by reading the titles, abstracts, and keywords to determine relevant studies (n = 130). A second screening was performed by reading the full text. Thirty-four full-text articles were studied. Finally, nine articles were selected for review and meta-analysis. The flow diagram is shown in Figure 2.

### 3.2. Characteristics of Included Studies

The nine studies involved 386 patients and investigated the effects of combining tDCS with RAT to improve the severity of upper limb deficits (n = 8), hand dexterity (n = 5), spasticity (n = 5), and activity (n= 2) in patients diagnosed with stroke. Table 1 shows the detailed information of each study.

In the included trials, the mean age of the participants was 65.80 years, and 178 (46.1%) of them were women. Most participants were subacute and chronic. Of all trials, 154 participants sustained the lesion in the left hemisphere, and 150 sustained it in the right hemisphere. Only one trial did not report the affected hemisphere or the type of stroke suffered by the participants. Of the 386 participants, 221 received a real tDCS, which could be anodic, cathodic, or dual tDCS, and 186 received a sham tDCS.

In all studies, the experimental intervention was robotic training for upper limb recovery combined with tDCS. Regarding tDCS, all studies stimulated motor area 1 (M1), corresponding to areas C3/C4 of systems 10–20. All trials used anodal stimulation (a-tDCS), except for one that additionally used cathodal stimulation (c-tDCS) in another experimental group [30]. All studies applied tDCS online, i.e., simultaneously with RAT, apart from Triccas (2015) [31], who worked with three groups, including one online, one offline pre-intervention (stimulation was applied prior to the robotic intervention), and one offline post-intervention (stimulation was applied after the robotic intervention). Edwars (2019) [32] applied offline pre-stimulation. Six of the articles included in the meta-analysis applied the intervention for 20 min, and Straudi (2016) [33] applied it for 30 min. Three studies performed the stimulation at an intensity of 1 mA [31,33,34]. The remaining studies carried out the stimulation at an intensity of 2 mA. All studies used saline and 35 cm^2^ electrodes. The data concerning the application of tDCS can be found in Table 2.

Regarding robotic intervention, it should be noted that only two studies used an exoskeleton [31,33], and the rest used an end-effector. Bimanual (n = 1) [35] and unimanual (n = 8) devices were used. All devices were distal, except one, which was proximal. The degrees of freedom (DoFs) of movement ranged from 2 to 3, but all were in relation to wrist motion. The DoFs used were abduction–adduction and flexion and pronosupination. The rest of the upper extremity was immobilised. Four studies did not report DoF-related information [32,33,34,36]. Three studies mobilised the entire limb through wrist movement in any of its three degrees of freedom [30,33,35]. Four studies fixed the upper limb and only allowed movements of the wrist [34,36,37,38]. Edwards (2019) [32] alternated movements of all of the upper limbs with wrist-only movements per day.

Only four studies provided data on the duration of the robotic treatment. The maximum time was 75 min [33], and the minimum time was 20 min [35]. Four studies did not provide information on the number of repetitions [31,32,34,35]. The maximum number of repetitions described was 1024 repetitions in one session [36,37,38], and the minimum was 400 repetitions [35]. All studies offered passive movements, i.e., movements that were performed by the robotic device. One studio performed self-passive movements. This was possible because the robotic device was bimanual; therefore, the patient himself was able to guide the movement of the affected upper limb thanks to the movement of the healthy upper limb [34]. Most studies (n = 4) used robotic-assisted movement, i.e., the patient initiated the movement, but it was the robot that completed the range of motion according to the DoF. Only three performed active movements; that is, the patient independently executed the entire process, but only if the participant wanted to perform it, i.e., it was not mandatory within the treatment. No studies reported problems with the use of robotic devices. All studies used visual feedback. One study did not report the type of task that was performed in the RAT [34]. Three studies used exergames [30,31,35], and five studies used visuomotor tasks [32,33,36,37,38]. The exergames had a more cognitive and playful component than the visuomotor tasks. Both exergames and visuomotor tasks were executed with the DoF offered by the robotic devices. Robotic devices had sensors and motors that defined the direction of movement and balanced the amount of force to complete the movement.

Information regarding RAT can be found in Table 3.

### 3.3. Quality

The quality of deficit severity, hand dexterity, spasticity, and activity was analysed in all of the studies. The heterogeneity was low. No study showed undirected evidence or imprecision. The publication bias was not clear in the deficit severity and hand dexterity, and there was no publication bias in spasticity nor activity. The DOI plot and funnel plot are available in Appendix A. The level of GRADE evidence is high. The data are available in Appendix A.

All studies were randomised clinical trials. The groups presented similar baseline characteristics and measurements, and the dropout rate was nonexistent or minimal. Allocation was concealed in all studies, and all studies blinded the participants and assessors. The intervener was not blinded in any of the studies. The mean score on the PEDro scale was 8 to 9, as presented in Appendix A.

#### Outcome Measures

The deficit severity measure was obtained from the FMA-UE scores (n = 8), including a general assessment of the upper limb and some shoulder, elbow, wrist, and hand measurements. The hand dexterity measures were obtained from the BBT scores (n = 5). The measures of spasticity were obtained using the MAS scale (n = 5), and the measures of activity were obtained using the BI (n = 2). The baseline results for the scores of the studies included are shown in Table 4.

### 3.4. Intervention Effects

A meta-analysis considering the intervention effect with its 95% confidence intervals was performed with the selected studies. The results did not show that the RAT intervention combined with tDCS for upper limb recovery after stroke is more beneficial than when RAT is combined with sham tDCS. The effect on deficit severity measured with the FMA-UE was examined in eight studies [31,33,34,35,36,37,38] involving a total of 354 patients. It was assessed with post-intervention measures. The global effect was −0.09 with a CI of (−0.31, 0.12). The results indicated that robotic training combined with tDCS did not improve the severity of the deficit compared to sham tDCS, as shown in Figure 3. The effect of RAT combined with tDCS compared with RAT alone on hand dexterity was examined in five studies [32,34,36,37,38] involving a total of 133 patients. The global effect was 0.12 with a CI of (−0.22, 0.46). The results indicated that the addition of tDCS to RAT does not improve hand dexterity, as shown in Figure 4. The effect of RAT combined with tDCS on spasticity was examined in the upper limb. Five studies evaluated spasticity but examined different structures [31,35,36,37,38]. The global effect was 0.04 with a CI of (−0.24, 0.32). No improvement in spasticity was found when combining RAT and tDCS, as shown in Figure 5. The effect of RAT combined with tDCS on activity was examined in two studies [31,35]. No improvement in activity was found when combining RAT and tDCS. The global effect was 0.66 with a CI of (−1.82, 3.14). The results are shown in Figure 6.

## 4. Discussion

The main objective of this review was to shed light on the effect of combining robot-assisted therapy (RAT) with transcranial direct current stimulation (tDCS) on the motor recovery of the upper limbs after a stroke. Additionally, the impact of this combined approach on spasticity, manual dexterity, and the performance of activities of daily living (ADLs) was investigated. However, no additive effect of tDCS and RAT was found in the recovery of upper limb motor function post-stroke, and there was no observed additive effect on spasticity, manual dexterity, or ADLs.

These results align with those of previous reviews [24,25,26,39] indicating that the effect of robot-assisted therapy (RAT) and transcranial direct current stimulation (tDCS) may differ in the lower limbs [24,39]. This difference could be attributed to the fact that the recovery of the upper limbs relies more on the integrity of the corticospinal pathway, whereas the recovery of the lower limbs depends on both this integrity and spinal structures that allow for the application of other techniques, such as transcutaneous spinal direct current stimulation [39,40]

RAT and tDCS have been combined with other modalities to achieve an additive effect in motor recovery after a stroke. Specifically, tDCS has been combined with mirror therapy and rehabilitative treatment, physiotherapy, and occupational therapy, showing positive effects on the motor recovery of the upper limbs and motor dexterity [41,42,43]. However, our results differ from those of these studies, which may be due to variability in tDCS protocols and patient characteristics. Various studies combining tDCS with rehabilitative treatment have yielded disparate results [44,45], underscoring the importance of variability in protocols and patient characteristics. Both studies address the combination of tDCS with rehabilitative treatment, but in one case, it is a c-tDCS protocol in chronic patients [45], and in the other case, it is an a-tDCS protocol in subacute patients [44]. On the other hand, the combination of tDCS and RAT may not adhere to certain recommendations for this combination to be more effective, specifically regarding the time between tDCS application and task execution, as well as the repetition of the same task [46]. Other authors have investigated ways to enhance the effects of RAT by combining it with techniques such as botulinum toxin or electrical stimulation [47,48], showing better results with the combination of these techniques. In contrast to the combination of RAT and tDCS, these techniques have positive effects on motor function and spasticity. This difference might be attributed to the plastic effects induced by RAT, i.e., the plastic effects produced by RAT may mask the effects of tDCS [39].

Furthermore, in RAT, the device used is important, as the variability of robotic devices can yield different outcomes [49]. At a mechanical level, a crucial distinction exists between exoskeletons and end-effectors [50]. Notably, the studies by Mazzoleni and Edward demonstrated substantial advantages in deficit severity [32,36,37,38], and both studies used end-effectors. Additional research underscores the advantages of end-effectors over exoskeletons [51]. End-effectors operate on a bottom-up model, focusing on simple movements to enhance functionality, which proves particularly effective after a stroke, especially in cases of severe deficits [52]. Moreover, end-effectors typically have a lower degree of freedom (DoF) compared to exoskeletons, aligning with the paradox of diminishing DoFs, suggesting that devices with fewer DoFs are more effective in severe motor impairment scenarios [53,54].

The dosage of movement is a critical factor in the recovery of patients diagnosed with stroke, with variables such as the extent of movement and the number of repetitions playing significant roles [55]. Although the optimal dose remains undetermined, emerging evidence suggests that a higher volume of movement is associated with better outcomes [56,57]. The results from the meta-analysis indicate a lack of adjuvant effects in robot-assisted therapy (RAT) and transcranial direct current stimulation (tDCS); however, studies with a higher number of replications and doses consistently achieve superior results [32,36,37,38]. The generalised lack of results may be attributed to factors such as completely passive movement, where the robotic device entirely dictates the movement. Interestingly, only three studies allowed patients to engage in voluntary active movements, emphasising the importance of active engagement, which increases cortical excitability and enhances plasticity [58,59]. The active role of the patient is a significant factor contributing to the success of RAT in motor function recovery [60,61]. Additionally, the frequency and total duration of the protocol are influential factors in determining the dose in RAT [62].

One way to enhance the effect of RAT would be to adapt the robotic device to the special characteristics of the pathology. Specifically, robotic devices have been designed for other pathologies, such as tendon disease, showing positive effects on recovery [14]. Another factor that improves the efficacy of RAT is the use of electrical impedance tomography to RAT, as this would help monitor the progress of patients and allow the treatment to be adapted to the specific needs of each person [63].

Following a stroke, the nervous system undergoes various changes, initiating a process of functional alterations known as brain plasticity [64]. Non-invasive Brain Stimulation (NIBS), particularly transcranial direct current stimulation (tDCS), could positively contribute to this plasticity [65]. The nervous system adopts different behaviours to cope with recovery after injury, with models such as the Vicariation Model and the interhemispheric competition model explaining the recovery process. While the Vicariation Model suggests that the unaffected hemisphere assumes the functions of the affected hemisphere, the interhemispheric competition model posits an inhibitory balance between hemispheres, with the unaffected hemisphere enhancing the inhibition of the affected hemisphere [66,67]. Techniques such as fractional anisotropy can aid in determining the appropriate type of NIBS, considering factors like cathodic or inhibitory stimulation in smaller lesions [68,69]. Despite the promising potential of NIBS in stroke recovery, the combination of tDCS with RAT presents controversy and a lack of conclusive results in this review. The variability in tDCS protocols, irrespective of the location, etiology, and lesion evolution, along with the presence or absence of Motor Evoked Potentials (MEPs), may contribute to the absence of consistent findings [67]. It is imperative for clinical evidence to guide healthcare professionals in applying the most appropriate techniques for individual patients, considering factors such as the time of evolution, predictive factors, and other relevant variables [70].

As previously mentioned, the effectiveness of RAT and tDCS may vary based on patient characteristics. Subsequent studies should explore device suitability tailored to specific patient profiles, including severity and potential for improvement. Notably, the studies included in this review did not select patients based on criteria predicting intervention efficacy. Limiting factors, such as advanced age, neglect, aphasia, or complete anterior circulation lesion (TACI), hinder the possibility of achieving favourable outcomes [50]. On the other hand, there are also predictive algorithms for the evolution of patients based on clinical scores and physiological data, for example, the specific use of scores, such as FMA-UE [71]. Despite classifications based on FMA-UE scores in studies such as that by Mazzoleni (2019) and the acknowledgment of the importance of presenting MEP in the study by Edwards et al. (2019), these criteria were not consistently utilised for treatment application or dose determination [32,38].

Overall, this review has shown that the combination of tDCS and RAT can have positive effects in certain groups, such as patients with subcortical lesions, but globally, no additional effects were found when combining RAT with tDCS. With the information from the studies, we can focus on the application of techniques based on the patients’ specific factors. We found that there are three variables that help generate profiles of the patients who can benefit the most from the combination of RAT and tDCS. These variables are as follows:The presence of MEP: Most studies suggest that patients with the presence of MEP, those who preserve the corticospinal pathway, benefit more from the effects of tDCS and RAT [32,33,34,40,58], although there are other studies that point in the opposite direction [30]. It is not clear which patients benefit the most from NIBS, and this may be an important variable in establishing this criterion, so further research is required.The severity of deficits: Most studies indicate that subjects with moderate deficits, measured with FMA-EU, benefit most from the combination of both techniques [31,34,38]. Baseline scores are predictors of the evolution of the subjects, so this information can help us decide whether or not a patient will benefit from the intervention [71].The time of evolution since stroke: The evidence in this regard is unclear, but it seems that patients in a subacute stage may benefit more [31,38]. However, there are studies indicating that chronic patients may also benefit [30,32,33,34].

Future research should take these variables into account and design more homogeneous studies that allow us to know more confidently which patients would benefit from the combination of RAT and tDCS.

In conclusion, based on the results of the meta-analysis and the analysis of the studies, the patients who can benefit the most from the techniques are those with a preserved, intact, or partially affected corticospinal pathway, with moderate deficits measured using the Fugl-Meyer Assessment, and those in a subacute stage.

The systematic review and meta-analysis presented in this study have limitations that may influence the results obtained. The data may be incomplete because most studies did not include a post-intervention follow-up; the high variability of tDCS and RAT protocols makes it difficult to compare results. In addition, the results of the meta-analysis may be influenced by the difference in the sample sizes of the studies. Finally, the study population was very heterogeneous, making it difficult to generalise the results.

## 5. Conclusions

The use of RAT combined with tDCS for the motor recovery of the upper limb after stroke is a promising rehabilitative technique, but there is some controversy about its effectiveness. Different studies yielded contradictory data about its usefulness, although there is some agreement that the patients who may benefit the most from its use are those with a preserved, intact, or partially affected corticospinal pathway with moderate motor deficits and those in a subacute stage.

However, further research is needed to more precisely define an optimal patient profile that can benefit from this technique to improve upper limb function and ADL performance and to establish more clear protocols for its application.

## Figures and Tables

**Figure 1 healthcare-12-00337-f001:**
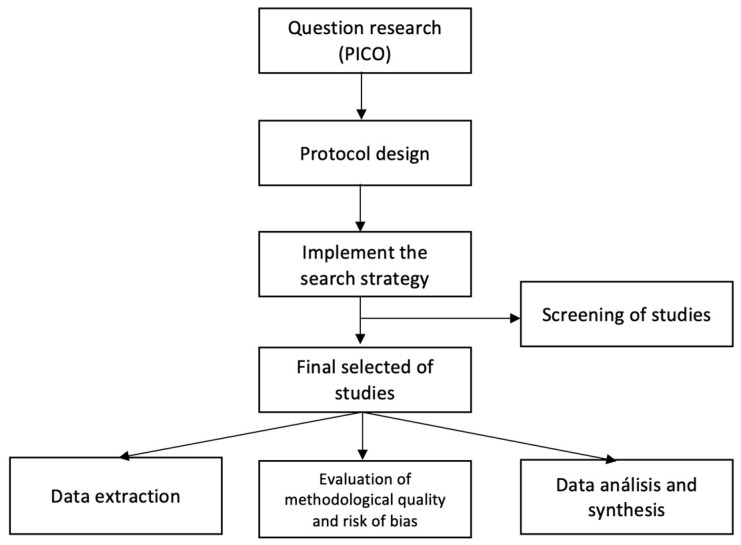
Methodology flow chart.

**Figure 2 healthcare-12-00337-f002:**
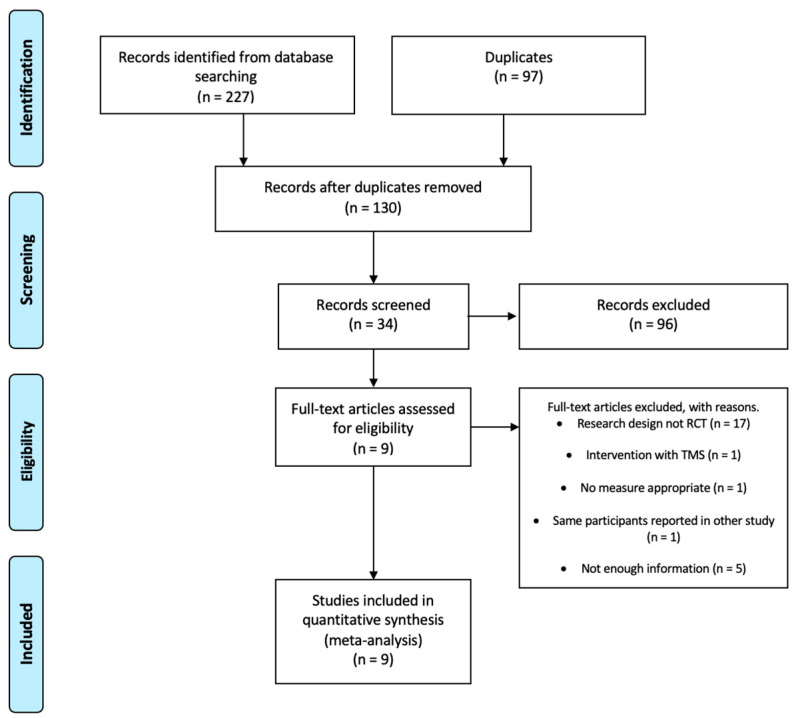
Flow of studies throughout the review.

**Figure 3 healthcare-12-00337-f003:**
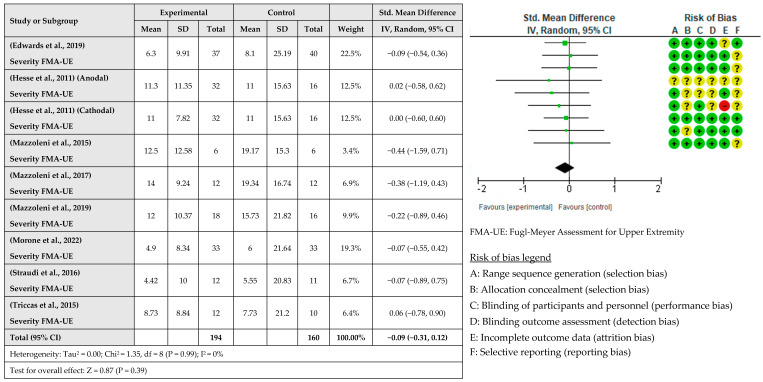
Meta-analysis of comparison between experimental group and control group results on the effect of deficit severity on stroke population [30,31,32,33,34,36,37,38]. The red circle (- sign) means high risk of bias, the yellow circle (? sign) means some concerns, and the green circle (+ sign) means low risk of bias.

**Figure 4 healthcare-12-00337-f004:**
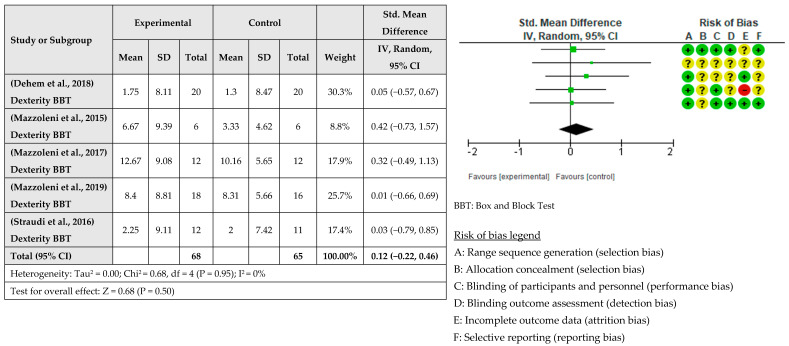
Meta-analysis of comparison between experimental group and control group results on the effect of hand dexterity on stroke population [33,35,36,37,38]. The red circle (- sign) means high risk of bias, the yellow circle (? sign) means some concerns, and the green circle (+ sign) means low risk of bias.

**Figure 5 healthcare-12-00337-f005:**
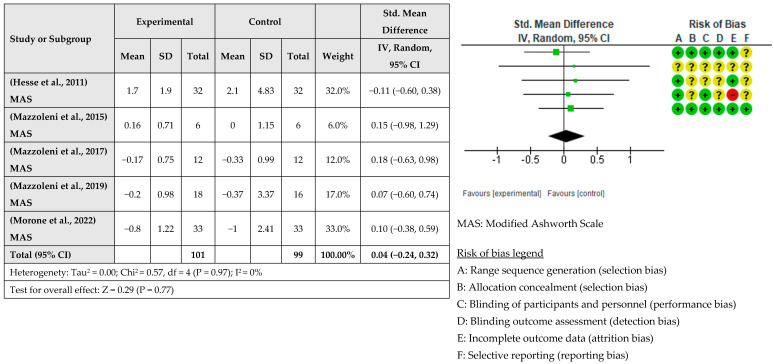
Meta-analysis of comparison between experimental group and control group results on the effect of spasticity on stroke population [30,34,36,37,38]. The red circle (- sign) means high risk of bias, the yellow circle (? sign) means some concerns, and the green circle (+ sign) means low risk of bias.

**Figure 6 healthcare-12-00337-f006:**
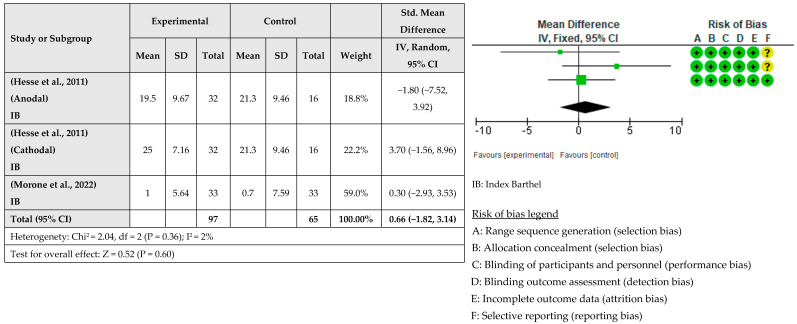
Meta-analysis of comparison between experimental group and control group results on the effect of activity on stroke population [30,34]. The yellow circle (? sign) means some concerns, and the green circle (+ sign) means low risk of bias.

**Table 1 healthcare-12-00337-t001:** Summary of included studies. Characteristics and designs of the included studies.

Study	Study Design	Time Poststroke	No. of Sessions/Week	Groups	Number of Patients	Diagnosis (Hemisphere Affected)	Age (Mean ± SD)	Outcomes
(Morone et al., 2022) [30]	RCT	Chronic	10 s/2 weeks	Group Ex: d-tDCS Group C: sham	66	36 RH 30 LH	Group Ex: 59.7 ± 12.5 Group C: 60.2 ± 16.1	FMA-UE; MAS; BBT; BI
(Triccas et al., 2015) [31]	RCT	Subacute and chronic	18 s/8 weeks	Group Ex: a-tDCS Group C: sham	23	11 LH 12 RH	Total Group: 63.4 ± 12	FMA
(Edwards et al., 2019) [32]	RCT	Chronic	36 s/12 weeks	Group Ex: a-tDCS Group C: sham	82	82 RH	See footnote *	FMA-UE
(Straudi et al., 2016) [33]	RCT	Subacute and chronic	10 s/2 weeks	Group Ex: d-tDCS Group C: sham	23	15 LH 8 RH	Group Ex: 52.7 ± 16 Group C: 64.3 ± 9.7	FMA; BBT
(Hesse et al., 2011) [34]	RCT	Subacute	30 s/6 weeks	Group Ex1: a-tDCS Group Ex2: c-tDCS Group C: sham	96	45 LH 51 RH	Group Ex1: 63.9 ± 10.5 Group Ex2: 65.4 ± 8.6 Group C: 65.6 ± 10.3	FMA-UE; MAS; BBT; BI
(Dehem et al., 2018) [35]	RCT; crossover	Chronic	2 s/1 weeks	Group Ex1: d-tDCS + ses sham Group Ex2: ses sham + d-tDCS	21	11 LH 10 RH	Group Ex: 62.73 ± 8 Group C: 58.1 ± 10.8	BBT
(Mazzoleni et al., 2015) [36]	RCT	Subacute	30 s/6 weeks	Group Ex: a-tDCS Group C: sham	12	4 RH 8 LH	Total Group: 75.9 ± 7	FMA-UE; MAS; BBT
(Mazzoleni et al., 2017) [37]	RCT	Subacute	30 s/6 weeks	Group Ex: a-tDCS Group C: sham	24	12 RH 12 LH	Group Ex: 70.0 ± 12.8 Group C: 75.25 ± 8.01	FMA-UE; MAS; BBT
(Mazzoleni et al., 2019) [38]	RCT	Subacute	30 s/6 weeks	Group Ex: a-tDCS Group C: sham	39	17 RH 22 LH	Group Ex: 67.5 ± 16.3 Group C: 68.74 ± 15.83	FMA-UE; MAS; BBT

RCT = randomised clinical trials; s = session; a-tDCS = anodal transcranial direct current stimulation; c-tDCS = cathodal transcranial direct current stimulation; d-tDCS = dual transcranial direct current stimulation; SD = standard deviation; Ex = experimental; Ex1 = experimental 1; Ex2 = experimental 2; C = control; LH = left hemispheric; RH = right hemispheric; FMA-UE = Fugl-Meyer Assessment for Upper Extremity; MAS = Modified Ashworth Scale; BBT: Box and Blocks Test; BI = Barthel Index. * Edward et al. (2019) [32] provided the median [interquartile range] of continuous variables: age, 70.0 [64.0, 77.0].

**Table 2 healthcare-12-00337-t002:** tDCS characteristics of the included studies.

Study	Stimulated Area	Timing	tDCS Time	Characteristics	Intensity
(Morone et al., 2022) [30]	D-tDCS M1	Online	20 min	Electrodes 35 cm^2^. Saline	2 mA
(Triccas et al., 2015) [31]	A-tDCS C3/C4 (M1) affected hemispheric	Online	20 min	Electrodes 35 cm^2^. Saline	1 mA
(Edwards et al., 2019) [32]	A-tDCS M1	Offline pre	20 min	Electrodes 35 cm^2^. Saline	2 mA
(Straudi et al., 2016) [33]	D-tDCS M1	Online	30 min	Electrodes 35 cm^2^. Saline	1 mA
(Hesse et al., 2011) [34]	A-tDCS: C3 C-tDCS: C3	Online	20 min	Electrodes 35 cm^2^. Saline	2 mA
(Dehem et al., 2018) [35]	D-tDCS	Online	20 min	Electrodes 35 cm^2^. Saline	1 mA
(Mazzoleni et al., 2015) [36]	A-tDCS M1	Online	20 min	Electrodes 35 cm^2^. Saline	2 mA
(Mazzoleni et al., 2017) [37]	A-tDCS M1	Online	20 min	Electrodes 35 cm^2^. Saline	2 mA
(Mazzoleni et al., 2019) [38]	A-tDCS M1	Online	20 min	Electrodes 35 cm^2^. Saline	2 mA

A-tDCS = anodal transcranial direct current stimulation; C-tDCS = cathodal transcranial direct current stimulation; D-tDCS = dual transcranial direct current stimulation.

**Table 3 healthcare-12-00337-t003:** RAT characteristics of the included studies.

Study	Robot Device	Robot Device (End-Effector vs. Exoskeleton)	Robot Device (Bimanual vs. Unimanual)	Robot Device (Distal vs. Proximal)	DOFs (Degrees of Freedom)	Robot-Assisted Training Time	Type of Task	No. of Repetitions
(Morone et al., 2022) [30]	Armeo Power II	Exoskeleton	Unimanual	Distal	6: elbow F; forearm S; wrist F; shoulder	40 min	Exergames	No data
(Triccas et al., 2015) [31]	Armeo Spring	Exoskeleton	Unimanual	Proximal	No data	75 min	Exergames	No data
(Edwards et al., 2019) [32]	MIT-Manus	End-effector	Unimanual	Distal	No data	60 min	Visuomotor task	1024 passives and assisted
(Straudi et al., 2016) [33]	REO Therapy System	End-effector	Unimanual	Distal	No data	30 min	Visuomotor task	No data
(Hesse et al., 2011) [34]	Bi-Manu Track	End-effector	Bimanual	Distal	2 F/E; P/S	20 min	No data	200 passives + 200 auto passives 800 total
(Dehem et al., 2018) [35]	REAplan robot	End-effector	Unimanual	Distal	No data	20 min	Exergames	No data
(Mazzoleni et al., 2015) [36]	InMotion	End-effector	Unimanual	Distal	3: F/E; P/S; ABD/ADD	No data	Visuomotor task	960 assisted + 16 passives 976 total
(Mazzoleni et al., 2017) [37]	InMotion	End-effector	Unimanual	Distal	3: F/E; P/S; ABD/ADD	No data	Visuomotor task	960 assisted + 16 passives 976 total
(Mazzoleni et al., 2019) [38]	InMotion	End-effector	Unimanual	Distal	3: F/E; P/S; ABD/ADD	No data	Visuomotor task	960 assisted + 16 passives 976 total

ABD/ADD = abduction/adduction; F/E = flexion/extension; P/S = pronation/supination.

**Table 4 healthcare-12-00337-t004:** Baseline outcomes of the included studies.

Study	FMA-UE (Mean ± SD)	BBT (Mean ± SD)	MAS (Mean ± SD)	BI (Mean ± SD)
(Morone et al., 2022) [30]	Group Ex: 25.8 ± 15.2 Group C: 30.7 ± 15.0	No data	Group Ex: 4.4 ± 2.3 Group C: 4.1 ± 1.7	Group Ex: 85.1 ± 11.0 Group C: 79.9 ± 14.0
(Triccas et al., 2015) [31]	Group Ex: 24.91 ± 16.01 Group C: 37.09 ± 13.57	No data	No data	No data
(Edwards et al., 2019) [32]	Group Ex: 25.7 ± 16.3 Group C: 25.3 ± 16.3	No data	No data	No data
(Straudi et al., 2016) [33]	Group Ex: 24.08 ± 16.6 Group C: 21.45 ± 13.23	Group Ex: 10.42 ± 15.47 Group C: 6.55 ± 11.67	No data	No data
(Hesse et al., 2011) [34]	Group Ex1: 7.8 ± 3.8 Group Ex2: 7.9 ± 3.4 Group C: 8.2 ± 4.4	Group Ex1: 0 Group Ex2: 0 Group C: 0	Group Ex1: 1.6 ± 2.9 Group Ex2: 1.0 ± 1.8 Group C: 1.4 ± 2.7	Group Ex1: 34.1 ± 6.4 Group Ex2: 34.2 ± 7.6 Group C: 35.0 ± 7.8
(Dehem et al., 2018) [35]	No data	Group R-S: 18.73 ± 13.3 Group S-R: 13.6 ± 14.3	No data	No data
(Mazzoleni et al., 2015) [36]	Group Ex: 28.00 ± 20.91 Group C: 41.83 ± 14.48	Group Ex: 11.33 ± 12.74 Group C: 20.5 ± 8.41	Group Ex: 0.67 ± 1.21 Group C: 0.33 ± 0.81	No data
(Mazzoleni et al., 2017) [37]	Group Ex: 37.33 ± 17.53 Group C: 37.83 ± 15.62	Group Ex: 15.00 ± 9.99 Group C: 15.42 ± 9.78	Group Ex: 0.75 ± 1.36 Group C: 0.50 ± 0.80	No data
(Mazzoleni et al., 2019) [38]	Group Ex: 34.20 ± 18.35 Group C: 34.11 ± 15.48	Group Ex: 15.95 ± 12.10 Group C: 12.32 ± 10.41	Group Ex: 1.1 ± 1.86 Group C: 1.58 ± 2.34	No data

BBT = Box and Block Test; C = Control; Ex = Experimental; Ex1 = Experimental a-tDCS; Ex2 = Experimental c-tDCS; FMA-UE = Fugl-Meyer Assessment for Upper Extremity; MAS = Modified Ashworth Scale; R-S = real tDCS and sham tDCS (crossover study); SD = standard deviation; S-R = sham tDCS and real tDCS (crossover study).

## Data Availability

The datasets used and/or analysed during the current study are available from the corresponding author upon reasonable request.

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
