# Peer review of "Is the Combination of Robot-Assisted Therapy and Transcranial Direct Current Stimulation Useful for Upper Limb Motor Recovery? A Systematic Review with Meta-Analysis"

_healthcare, 2024, doi:10.3390/healthcare12030337_

Round 1
Reviewer 1 Report
Comments and Suggestions for Authors
The primary objective of the research was to determine the effectiveness of combining robot-assisted therapy (RAT) and transcranial direct current stimulation (tDCS) in upper limb motor recovery after stroke​​. The topic appears to be original and relevant in the field, addressing the need for effective rehabilitation strategies for stroke survivors, a significant cause of disability worldwide. The study aims to evaluate the combined use of two promising approaches, RAT and tDCS, for enhancing motor recovery, which could fill a gap in stroke rehabilitation methods​​.
-Authors should consider a broader range of studies, including non-randomized trials.
- They should also focus on the individualized application of tDCS and RAT based on patient-specific factor
-Some papers should be included to use different methods to evaluate and analyse the mechanisms in the rehabilitation process:
. Picelli, A., Chemello, E., Castellazzi, P., Roncari, L., Waldner, A., Saltuari, L., & Smania, N. (2015). Combined effects of transcranial direct current stimulation (tDCS) and transcutaneous spinal direct current stimulation (tsDCS) on robot-assisted gait training in patients with chronic stroke: a pilot, double blind, randomized controlled trial. Restorative Neurology and Neuroscience, 33(3), 357-368.
. Niestanak, V. D., Moghaddam, M. M. (2017, October). A new underactuated mechanism of hand tendon injury rehabilitation. In 2017 5th RSI International Conference on Robotics and Mechatronics (ICRoM) (pp. 400-405). IEEE.
. Naro, A., & Calabrò, R. S. (2022). Improving Upper Limb and Gait Rehabilitation Outcomes in Post-Stroke Patients: A Scoping Review on the Additional Effects of Non-Invasive Brain Stimulation When Combined with Robot-Aided Rehabilitation. Brain Sciences, 12(11), 1511.
. Moshaei, A., Chinnakkonda Ravi, A. K., & Kern, T. A. (2023). Development of a New Control System for a Rehabilitation Robot Using Electrical Impedance Tomography and Artificial Intelligence. Biomimetics, 8(5), 420.
Reviewer 2 Report
Comments and Suggestions for Authors
Stroke is the third leading cause of disability in the world, and effective rehabilitation is needed to improve lost functionality.
Robot-assisted therapy (RAT) and transcranial direct current stimulation (tDCS) are promising approaches for stroke survivors.
The objective of the authors is to determine the effectiveness of the combined use of RAT and tDCS in upper limb motor recovery after stroke.
They performed a systematic review with meta-analysis of randomized trials.
They included Nine studies were included and analysed in the meta-analysis.
Outcomes analysed were deficit severity, hand dexterity, spasticity and activity.
Authors concluded that: (a) there is no evidence of an additional effect when adding tDCS to the RAT for upper limb recovery after stroke. (b) tDCS combined with RAT does not improve upper limb motor function, spasticity and/or hand dexterity. (c) Future research should focus on the use of RAT protocols in which the patient is given an active role, focusing on intensity and dosage, and determining how certain variables influence the success of RAT.
The study touches a hot topic
Some improvement are needed
1 The abstract seems fragmented. It seems like a list of statements. Please smooth and modify the flow and the sentences
2. Also the introduction (probably for the revisions) must be smoothed. Please use “[]” for the citations
3. The design of the methodology is unclear in some passages. Add a flow chart. Add a brief introduction of the methodology before the paragraphs. There are dozins of paragraphs, some are with one or two sentence. A reader could lost in them. Please rewrite.
4. Reduce the number paragraphs also in the results, that however, seem effective.
5. Discussion must report both the limitations encountered in the studies and the limitation of the review.
6. Probably there is a comparison with other studies in the discussion, but I am a bit lost..
7. Reference are not correctly cited, sometimes you sue the incorrect () with inside the number, sometimes you use the ucorrect () with the date. Use the standard.
8 The conclusions seem a list of notes.
Round 2
Reviewer 2 Report
Comments and Suggestions for Authors
NA